# Research on Output Characteristics of Microscale BST Laminate Structure Based on Mixed Finite Element Method

**DOI:** 10.3390/mi14040755

**Published:** 2023-03-29

**Authors:** Ying Luo, Tian Pu, Hongguang Liu

**Affiliations:** 1Faculty of Civil Engineering and Mechanics, Jiangsu University, Zhenjiang 212013, China; 2National Center for International Research on Structural Health Management of Critical Components, Jiangsu University, Zhenjiang 212013, China

**Keywords:** flexoelectric effect, micro-scale electromechanical coupling characteristics, C^1^ continuous mixed element, cross-scale laminated structure

## Abstract

The flexoelectric effect, which is sensitive to size, refers to the phenomenon of coupling between the strain gradient and electrical polarization and involves higher-order derivatives of physical quantities such as displacement, and the analytical process is complicated and difficult. Therefore, in this paper, a mixed finite element method is developed considering the effects of size effect and flexoelectric effect on the electromechanical coupling behavior of microscale flexoelectric materials. Based on the theoretical model of enthalpy density and the modified couple stress theory, the theoretical model and finite element model of microscale flexoelectric effect are established, and the Lagrange multiplier is used to coordinate the higher-order derivative relationship between the displacement field and its gradient, and the C^1^ continuous quadrilateral 8-node (displacement and potential) and 4-node (displacement gradient and Lagrange multipliers) flexoelectric mixed element. By comparing the numerical calculation results and analytical solutions of the electrical output characteristics of the microscale BST/PDMS laminated cantilever structure, it is proved that the mixed finite element method designed in this paper is an effective tool for studying the electromechanical coupling behavior of flexoelectric materials.

## 1. Introduction

In the 1960s, people discovered the non-traditional “piezoelectric behavior” in non-uniformly deformed materials—the flexoelectric effect, that is, the electrical polarization of the dielectric material induced by the strain gradient change (positive flexoelectric effect) and caused by the electric field gradient. It produces a phenomenon of strain response (inverse flexoelectric effect) [1]. Different from piezoelectric materials, flexoelectric materials do not require pre-polarization, have no temperature limit, and have scale sensitivity. They have wide application prospects in the fields of micro- and nano-scale energy harvesting, sensing and driving, and have attracted much attention from researchers [2]. Some achievements have been made in the preparation of flexoelectric materials [3], performance testing and characterization [4], theoretical modeling of electromechanical coupling effects [5], numerical simulation methods [6], and application exploration [7,8].

The non-locality of the strain gradient or polarization (electric field) gradient makes the boundary value problem with respect to the flexoelectric effect of a solid dielectric material dominated by a fourth-order partial differential equation, which is only applicable when models such as one-dimensional or axisymmetric models (Simple geometric problems such as cylinders, disks or plates) can be solved analytically, while the flexoelectric response problems of those complex structures need to be solved by corresponding numerical methods. It is only in the process of numerical solution that the higher-order partial differential equations require that each element meet the higher-order continuity condition; that is, the shape function not only needs to satisfy the function continuity on the edge of the element but also the first-order derivative function of the function needs to satisfy the continuity function.

Shu et al. [9] established a mixed 2D finite element with displacement gradients and Lagrangian multipliers as extra-nodal degrees of freedom to analyze the electrical polarization generated in a single crystal flexoelectric cantilever beam and used the C^0^ continuous shape. The function achieves the same convergence as the C^1^ continuous function. Amanatidou and Aravas [10] developed some mixed elements, which can obtain relatively accurate solutions in some classical problems (such as single-crystal cantilever beams), but the theory requires additional material intrinsic parameters that are difficult to determine experimentally. Couple stress theory [11] is a method for studying microscale effects, which has the advantage of having few additional material intrinsic constants and is relatively easy to determine experimentally. Regarding the general strain gradient elasticity [12] problem, Hutchinson [13] and Shu [14] took the couple-stress rotation angle as an additional nodal degree of freedom and solved the flexoelectric higher-order partial differential equation by the mixed finite element method, which verifies that the couple-stress Feasibility of theoretically dealing with flexoelectric problems.

Aiming at the high-order continuity problem encountered in the numerical analysis of flexoelectric response, Abdollahi et al. [15] first used the Local Maximum Entropy (LME) meshless method to try to solve the modeling of flexoelectric single crystal cantilever structure. In order to simplify the model and reduce the difficulty of modeling, it deliberately ignores the influence of part of the flexoelectric effect on the boundary value problem. Based on the work of Amanatidou et al., Mao et al. [16] constructed a mixed formulation of flexoelectricity with displacement and displacement gradient as independent variables and developed a 2D cell to model BVPs (boundary value problems). However, the model they developed has an additional degree of freedom of the polarization node, which still has many problems that the constants are difficult to determine. Deng et al. [17] proposed a mixed finite element method with displacement, electric potential, displacement gradient and Lagrange multipliers as degrees of freedom to study the flexoelectric coupling response of dielectric materials and found that after considering the strain gradient, it can make the strain field smoother near the boundary. Nguyen et al. [18] used the IGA (Isogeometric analysis) method to solve the high-order continuum problem of the flexoelectric effect and focused on the influence of the flexoelectric effect on the potential of the nanowire, but the research on the scale effect at the micro-nano scale was not comprehensive enough. The above work has well promoted the numerical modeling research of flexoelectric effect, but the influence of size effect is ignored in the above research, and the size effect of flexoelectric materials is only studied based on the strain gradient theory, resulting in many parameters that are difficult to determine. In fact, with the continuous development of modern processing technology, the structure of micron-scale MEMS functional devices in practical applications is constantly emerging, and the preparation process is becoming more and more mature so that researchers must refocus on the research of micro-scale flexoelectric effect. BST ceramics are the preferred dielectric materials for the preparation of flexoelectric functional structures (the flexoelectric coefficient is much higher than other dielectric materials), and combined with flexible materials (such as PDMS) to form a flexoelectric composite/laminate structure, it can give full play to its power-to-electricity conversion performance can effectively avoid the disadvantage that the micro-scale structure is difficult to form and damaged.

In this paper, the modified couple stress theory is introduced into the flexoelectric enthalpy density theoretical model, and a mixed finite element method is developed to explore the flexoelectric effect by taking the cantilever structure laminated by microscale ceramic material BST and flexible material PDMS as an example. And the influence of scale effect on the output voltage of microscale flexoelectric laminate structure and the energy harvesting efficiency of cross-scale flexoelectric structure can be improved through the parametric design of the structure.

## 2. Electrical Enthalpy Density Theoretical Model of Microscale Flexoelectric Effect

Shen and Hu established the enthalpy density theory to describe the mechanical energy density, dielectric energy density, piezoelectric energy density and flexural electrical energy density of dielectric elastomers according to the energy method [19,20], and the specific form is as follows:(1)$$\begin{array}{c} G_{b}=\frac{1}{2}C_{ijkl}\varepsilon_{ij}\varepsilon_{kl}-\frac{1}{2}k_{ij}E_{i}E_{j}-e_{ijk}E_{i}\varepsilon_{jk}\\ \;\;\;-g_{ijkl}E_{k}\varepsilon_{ij,l}-f_{klij}\varepsilon_{ij}E_{k,l}-\frac{1}{2}b_{ijkl}E_{i,j}E_{k,l} \end{array}$$
where $C_{ijkl}$ is the elastic coefficient tensor, $k_{ij}$ is the dielectric constant, $\varepsilon_{jk}$ is the strain tensor, $E_{k}$ is the electric field vector, $e_{ijk}$ is the piezoelectric constant tensor, $g_{ijkl}$, $f_{klij}$ are the positive and inverse flexoelectric coefficient tensors, respectively, $\varepsilon_{ij,l}$ are the strain gradient tensors, are the $b_{ijkl}$ is the non-local electric field coupling coefficient tensor.

In order to accurately characterize the scale characteristics of dielectric solid materials at the microscale, a microscale factor term $C_{ijkl}^{0}\chi_{ij}\chi_{kl}/2$ is added on the premise of mutual verification with the experimental results, and the positive and inverse flexoelectric coefficients of the dielectric materials are considered objectively $g_{ijkl}=-f_{ijkl}$. Then a new enthalpy density function is obtained:(2)$$\begin{array}{c} \;G_{b}=\frac{1}{2}C_{ijkl}\varepsilon_{ij}\varepsilon_{kl}+\frac{1}{2}C_{ijkl}^{0}\chi_{ij}\chi_{kl}-\frac{1}{2}k_{ij}E_{i}E_{j}-e_{ijk}E_{i}\varepsilon_{jk}\\ -g_{ijkl}\varepsilon_{ij}E_{k,l}-f_{klij}\varepsilon_{kl}E_{i,j}-\frac{1}{2}b_{ijkl}E_{i,j}E_{k,l} \end{array}$$

Among them, $C_{ijkl}^{0}=Gl^{2}$ is the microscopic elastic stiffness matrix, l is the intrinsic constant of the material, which only depends on the microstructural characteristics of the material and is related to the microscale properties of the material, but has nothing to do with the specific structural size of the material, G is the Lame constant; the couple stress term $\chi_{ij}$ represents that as the relevant scale gradually decreases, the ion polarization effect increases, which in turn affects the enthalpy density of the material; at the same time, the contribution of the orientation polarization effect, which accounts for a large proportion in the macroscopic scale, to the structural, electric enthalpy density decreases with the decrease of the scale, which is embodied in the change of the macroscopic strain term $\varepsilon_{ij}$.

Studies have shown that there is no piezoelectric effect in the centrosymmetric crystal point group dielectric solid materials, which can be made according to the stress tensor $\sigma_{ij}$, high-order stress tensor $\tau_{jkl}$, electric displacement vector $D_{i}$, couple stress tensor $m_{ij}$, electric quadrupole moment $Q_{ij}$ and electric enthalpy density function [21], the micro-scale flexural-electrical coupling constitutive equation with only flexoelectric effect is obtained:(3)$$\left\{ \begin{array}{l} \sigma_{ij}=C_{ijkl}\varepsilon_{kl}/2\\ \tau_{jkl}=-\mu_{ijkl}E_{i}\\ m_{ij}=C_{ijkl}^{0}\chi_{kl}/2=Gl^{2}\delta_{ik}\delta_{jl}\chi_{kl}/2\\ D_{i}=k_{ij}E_{j}+\mu_{ijkl}\varepsilon_{jk,l}\\ Q_{ij}=b_{ijkl}E_{k,l} \end{array}\right.$$

## 3. Constrained Variation Principle

For a two-dimensional plane problem, the curvature tensor and couple stress tensor are:(4)$$\left\{ \begin{array}{l} \chi =\left(\begin{array}{l} \chi_{13}\\ \chi_{23} \end{array}\right)=\frac{1}{2}\left(\nabla_{1}\theta +{\left(\nabla_{1}\theta \right)}^{T}\right)\\ m=2Gl^{2}\chi =2Gl^{2}\left(\begin{array}{l} \chi_{13}\\ \chi_{23} \end{array}\right) \end{array}\right.$$
where rotation vector $\theta_{1}=\theta_{2}=0$, $\theta_{3}=\frac{1}{2}\left(\frac{\partial v}{\partial x}-\frac{\partial u}{\partial y}\right)$.

On the boundary it should satisfy:(5)$$u=\bar{u},\;\mathrm{on}\;\Gamma_{u};\;\theta =\bar{\theta },\;\mathrm{on}\;\Gamma_{\theta };\;\varphi =\bar{\varphi },\;\mathrm{on}\;\Gamma_{\varphi }$$

Integrate the inner term and boundary term of the constitutive relation of microscale flexoelectric materials, and integrate $E=-\nabla_{1}\varphi$ and $\chi =\nabla_{1}\theta$, the enthalpy function of the two-dimensional enthalpy model can be obtained:(6)$$\begin{array}{cc} \Gamma = & \frac{1}{2}\int_{\Omega }{\left(C\nabla_{3}u\right)}^{T}\nabla_{3}udV+\frac{1}{2}\int_{\Omega }{\left(C_{0}\nabla_{1}\theta \right)}^{T}\nabla_{1}\theta dV\\ & +\frac{1}{2}\int_{\Omega }{\left(\mu^{T}\nabla_{1}\varphi \right)}^{T}\nabla_{2}\varepsilon dV+\frac{1}{2}\int_{\Omega }{\left(\mu \nabla_{2}\varepsilon -\kappa \nabla_{1}\varphi \right)}^{T}\nabla_{1}\varphi dV\\ & -\int_{\partial \Omega }\left(p^{T}u+d^{T}\theta \right)dS-\int_{\partial \Omega }b\varphi dS \end{array}$$
where p is the surface force vector, d is the surface force couple, and b is the charge surface density.

For a two-dimensional plane problem, the displacement gradient vector can be expressed as:(7)$$\begin{array}{c} \psi ={\left(\psi_{11}\;\psi_{12}\;\psi_{21}\;\psi_{22}\right)}^{T}={\left(u_{1,1}\;u_{1,2}\;u_{2,1}\;u_{2,2}\right)}^{T}\\ \;=L_{\psi u}\left(\begin{array}{l} u\\ v \end{array}\right)=\nabla_{4}u \end{array}$$

Then the corresponding strain, strain gradient, rotation angle and rotation gradient can be expressed as a functional form with displacement gradient:(8)$$\left\{ \begin{array}{l} \varepsilon =L_{\varepsilon \psi }\psi \\ \eta =\nabla_{3}\varepsilon =\nabla_{3}L_{\varepsilon \psi }\psi =L_{\eta \varepsilon }L_{\varepsilon \psi }\psi =L_{\eta \psi }\psi \\ \theta =\theta_{3}=\left(\psi_{21}-\psi_{12}\right)/2=L_{\theta \psi }\psi \\ \chi ={\left(2\chi_{13}\;2\chi_{23}\right)}^{T}=\nabla_{1}\theta =\nabla_{1}L_{\theta \psi }\psi =L_{\chi \psi }\psi \end{array}\right.$$

Substitute Equation (8) into Equation (6) and introduce the constraint relationship between displacement and displacement gradient Equation (7), set the Lagrange multiplier to be α, then the induction enthalpy function becomes:(9)$$\begin{array}{c} \Gamma =\frac{1}{2}\int_{\Omega }{\left(C\nabla_{3}u\right)}^{T}\nabla_{3}udV+\frac{1}{2}\int_{\Omega }{\left(C_{0}\nabla_{1}L_{\theta \psi }\psi \right)}^{T}\nabla_{1}L_{\theta \psi }\psi dV\\ \;+\frac{1}{2}\int_{\Omega }{\left(\mu^{T}\nabla_{1}\varphi \right)}^{T}L_{\varepsilon \psi }\psi dV+\frac{1}{2}\int_{\Omega }{\left(\mu L_{\eta \psi }\psi -\kappa \nabla_{1}\varphi \right)}^{T}\nabla_{1}\varphi dV\\ \;-\int_{\partial \Omega }\left(p^{T}u+d^{T}\theta \right)dS-\int_{\partial \Omega }b\varphi dS \end{array}$$

The formula obtained by dividing the enthalpy density function after the variation is valid for any δu, δφ, δψ and δα, then we can get:(10)$$\left\{ \begin{array}{l} \nabla_{3}^{T}C\nabla_{3}u-\nabla_{4}^{T}\alpha =0\\ \nabla_{1}^{T}\mu L_{\eta \psi }\psi -\nabla_{1}^{T}\kappa^{T}\nabla_{1}\varphi =0\\ L_{\eta \psi }^{T}\mu^{T}\nabla_{1}\varphi +L_{\chi \psi }^{T}C_{0}L_{\chi \psi }\psi +\alpha =0\\ -\nabla_{4}u+\psi =0 \end{array}\right.$$

The boundary terms must satisfy:(11)$$\left\{ \begin{array}{l} n_{1}\cdot \left(C\nabla_{3}u\right)-n_{4}\cdot \alpha =p\\ n_{2}\cdot \left(\mu L_{\eta \psi }\psi -\kappa \nabla_{1}\varphi \right)=b\\ n_{3}\cdot \mu^{T}\nabla_{1}\varphi =0\\ n_{4}\cdot \left(C_{0}L_{\chi \psi }\psi \right)=dL_{\theta \psi }^{T} \end{array}\right.$$

Equation (11) is the Navier equation of the flexoelectric effect in the microscale flexoelectric structure, and its boundary conditions are:(12)$$u=\;\bar{u},\;\mathrm{on}\;u;\varphi =\;\bar{\varphi },\;\mathrm{on}\;\varphi ;\psi =\;\bar{\psi },\;\mathrm{on}\;\psi$$
where $\;\bar{u}$, $\;\bar{\varphi }$ and $\;\bar{\psi }$ are the known displacement, potential and displacement gradient on the boundary.

For the BST material of the m3m point group, the forward flexoelectric coefficient matrix is:(13)$$\mu =\left[\begin{array}{llllll} \mu_{11} & \;0\; & \mu_{14} & \;0\; & \mu_{111} & \;0\\ 0\; & \mu_{14} & \;0\; & \mu_{11} & \;0\; & \mu_{111} \end{array}\right]$$

For the isotropic m3m point group crystalline material, the elastic coefficient matrix $C_{0}$ is:(14)$$C_{0}=Gl^{2}\left[\begin{array}{cc} 1 & 0\\ 1 & 0 \end{array}\right]$$

The dielectric coefficient matrix κ, and the specific form of the operator tensor that appears in the calculation process is as follows:(15)$$\begin{array}{l} \nabla_{1}=\left[\begin{array}{l} \frac{\partial }{\partial x}\\ \frac{\partial }{\partial y} \end{array}\right],\;\nabla_{2}=\left[\begin{array}{lll} \nabla_{1} & & \\ & \nabla_{1} & \\ & & \nabla_{1} \end{array}\right],\;\nabla_{3}=\left[\begin{array}{ll} \;\frac{\partial }{\partial x} & \;0\\ \;0\; & \frac{\partial }{\partial x}\;\\ \;\frac{\partial }{\partial y} & \frac{\partial }{\partial x} \end{array}\right],\;\nabla_{4}=\left[\begin{array}{ll} \nabla_{1} & \\ & \nabla_{1} \end{array}\right],\;\\ L_{\varepsilon \psi }=\left[\begin{array}{cccc} 1 & 0 & 0 & 0\\ 0 & 0 & 0 & 1\\ 0 & 0.5 & 0.5 & 0 \end{array}\right],\;L_{\theta \psi }=[0\;-0.5\;0.5\;0],\;\\ L_{\chi \psi }=\nabla_{1}L_{\theta \psi },\;L_{\eta \psi }=\nabla_{3}L_{\varepsilon \psi } \end{array}$$

## 4. Microscale Flexoelectric Mixed Cell Construction

Figure 1 shows the mixed element of quadrilateral four-nodes and quadrilateral eight-nodes selected in this paper.

The displacement is 8-node interpolation, and each element node has two degrees of freedom:(16)$$u=\left(\begin{array}{l} u\\ v \end{array}\right)=\sum_{i=1}^{8}\left[\begin{array}{ll} N_{i}^{\left(8\right)} & \;0\\ 0\; & N_{i}^{\left(8\right)} \end{array}\right]\left(\begin{array}{l} u_{i}\\ v_{i} \end{array}\right)=N_{u}\;\tilde{u}$$
where $N_{i}^{\left(8\right)}$ represents the 8-node shape function.

The potential also selects an 8-node interpolation function, and each element node has 1 degree of freedom:(17)$$\varphi =\sum_{i=1}^{8}N_{{}_{i}}^{\left(8\right)}\varphi_{i}=N_{\varphi }\;\tilde{\varphi }$$

The displacement gradient selects 4-node interpolation, and each element node has 4 degrees of freedom:(18)$$\psi =\sum_{i=1}^{4}\left[\begin{array}{llll} N_{{}_{i}}^{\left(4\right)} & & & \\ & N_{{}_{i}}^{\left(4\right)} & & \\ & & N_{{}_{i}}^{\left(4\right)} & \\ & & & N_{{}_{i}}^{\left(4\right)} \end{array}\right]\left(\begin{array}{l} \psi_{11i}\\ \psi_{12i}\\ \psi_{21i}\\ \psi_{22i} \end{array}\right)=N_{\psi }\;\tilde{\psi }$$
where $N_{i}^{\left(4\right)}$ denotes a 4-node shape function.

The Lagrange multiplier selects 4-node interpolation, and each element node has 4 degrees of freedom:(19)$$\alpha =\sum_{i=1}^{4}\left[\begin{array}{llll} N_{{}_{i}}^{\left(4\right)} & & & \\ & N_{{}_{i}}^{\left(4\right)} & & \\ & & N_{{}_{i}}^{\left(4\right)} & \\ & & & N_{{}_{i}}^{\left(4\right)} \end{array}\right]\left(\begin{array}{l} \alpha_{11i}\\ \alpha_{12i}\\ \alpha_{21i}\\ \alpha_{22i} \end{array}\right)=N_{\alpha }\;\tilde{\alpha }$$

The constraints between the corresponding basic variables are:(20)$$\left\{ \begin{array}{l} \varepsilon =L_{\varepsilon u}N_{u}\;\tilde{u}=B_{\varepsilon }\;\tilde{u}\\ E=L_{E\varphi }\varphi =L_{E\varphi }N_{\varphi }\;\tilde{\varphi }=B_{E}\;\tilde{\varphi }\\ \eta =L_{\eta \psi }\psi =L_{\eta \psi }N_{\psi }\;\tilde{\psi }=B_{\eta }\;\tilde{\psi }\\ \psi =L_{\psi u}u=L_{\psi u}N_{u}\;\tilde{u}=B_{\bar{u}}\;\tilde{u}\\ \chi =\nabla_{1}\theta =\nabla_{1}L_{\theta \psi }\psi =\nabla_{1}L_{\theta \psi }N_{\psi }\;\tilde{\psi }=B_{\chi }\;\tilde{\psi } \end{array}\right.$$

According to Equations (13)–(20), it can be deduced that the equivalent strain matrices $B_{\varepsilon }$, $B_{E}$, $B_{\eta }$, $B_{\bar{u}}$ and $B_{\chi }$ respectively are:(21)$$\left\{ \begin{array}{l} B_{\varepsilon }=L_{\varepsilon u}N_{u}\\ B_{E}=L_{E\varphi }N_{\varphi }\\ B_{\eta }=L_{\eta \psi }N_{\psi }\\ B_{\bar{u}}=L_{\psi u}N_{u}\\ B_{\chi }=\nabla_{1}L_{\theta \psi }N_{\psi } \end{array}\right.$$

## 5. Numerical Simulation and Performance Analysis of Microscale Flexoelectric Laminated Beams

### 5.1. Output Voltage Analysis of Single Functional Layer BST/PDMS Laminated Structure

On the basis of the previous section of this article, the mechanical and electrical coupling behavior of the micro-scale flexoelectric laminated cantilever beam with a flexible layer attached to the surface is explored by following the constitutive relationship of the micro-scale flexoelectric material. The PDMS flexible layer attached to the beam surface can effectively reduce the fabrication difficulty of micro-scale flexoelectric composite beams and play a protective role in static and dynamic tests.

Figure 2 shows a finite element schematic diagram of a microscale flexoelectric laminated cantilever beam structure with a fixed left end and a free right end. The cross-section of the laminated beam is rectangular, with a BST flexoelectric layer in the middle, and its upper and lower surfaces are coated with a high-temperature electrode layer with a negligible thickness, and a PDMS flexible layer is attached to the outside of the electrode layer. The beam length is 2 cm, the BST layer thickness is $h_{B}$, and the flexible layer thickness is both $h_{P}/2=5\;\mu m$, intrinsic size characteristic parameters of the material is l=5 μm. The material parameters of the laminated beams are shown in Table 1.

The flexoelectric laminated beams act respectively as follows: (1) the lateral concentrated force at the free end F=−100 μN, (2) the distributed load on the upper surface q=−10 μN/m, and (3) the force couple acting on the free end of the laminated beam $M=10\times 10^{-5}\mathrm{Nm}$, and the displacement between the structural layer elements is used as the constraint boundary condition.

The upper and lower surfaces of the BST sensitive layer are equipotential surfaces. Figure 3 shows the variation of the output voltage of the microscale flexoelectric laminated beam with the thickness of the BST layer under different static loads calculated by the mixed finite element method and the analytical method, where the voltage represents the output voltage in the open-circuit environment. As can be seen from the figure:

① The agreement between the analytical solution and the numerical solution is relatively high. The force couple, uniform load and concentrated force reach their peaks at the thickness of the BST layer of 4.3 μm. At this time, the errors are 0.7%, 0.5% and 1.1%, respectively, which verifies the finite element method. accuracy and universality;

② The variation trend of the voltage output of the laminate structure under the three loads remains the same. The thickness of PDMS remains unchanged, and the output voltage of the BST/PDMS laminate structure first increases and then decreases with the increase in the thickness of the BST layer. This is due to the combined effect of the increase in the thickness of the BST, the increase in the specific gravity of the mechanoelectrical conversion functional layer of the structure and the decrease in the size effect (strain gradient).

### 5.2. Analysis of Output Voltage of Multilayer BST Laminated Structure

Figure 4 shows a schematic diagram of the three-layer BST/PDMS lamp-beam energy harvesting device. The thickness of BST layer is taken as $h_{B}$, the thickness of the middle two layers of PDMS layer are $h_{P}=20\;\mu m$, the thickness of the upper and lower surfaces of PDMS layer are $h_{Pu}=0.5h_{Pm}$, and the intrinsic size characteristic parameters of the material is l=5 μm. The grid division is the same as in Section 5.1, and the thickness of the electrode layer on the upper and lower surfaces of BST is ignored. BST layers are connected in series, and the access circuit is electrically open. A uniform load q=−100 μN/m is applied to the upper surface of each BST layer of the laminated beam.

From the variation law of the output voltage of the structure with BST thickness shown in Figure 5, it can be seen that the numerical solution is in good agreement with the analytical solution; as the thickness of the BST layer increases, the output voltage of the overall structure increases first and then decreases; The output voltage of the composite beam structure reaches the maximum value when the thickness of the BST layer is 3 μm, and the error between the numerical solution and the analytical solution is about 0.7%, which is within the acceptable range.

Setting the thickness of the BST layer (3 μm) and the uniform load q=−100 μN/m applied to each BST layer, the variation law of the output voltage of the BST/PDMS flexoelectric laminated cantilever structure with the number of BST layers under different PDMS layer thicknesses is shown in Figure 6.

It can be seen from Figure 6 that when the thickness of the PDMS layer is constant, the output voltage of each BST layer transducer element decreases with the increase in the number of layers of the microscale flexoelectric laminate structure; when the thickness of the PDMS layer is 20 μm (When the thickness of the BST layer is about 5 times), the output voltage of the structure decreases slightly with the increase in the number of BST layers. This is because the increase in the number of BST flexoelectric functional layers leads to an increase in the overall stiffness of the BST/PDMS laminate structure, which in turn leads to a decrease in the output voltage of each BST transducer element.

Selected $h_{P}=5h_{B}$, the output voltage of the BST/PDMS laminate structure with different layers of BST sensitive layers under the action of uniform load is shown in Figure 7. It can be seen that the output voltage of the BST/PDMS laminated structure with different BST layers increases first and then decreases with the increase in the thickness of the BST layer, and the variation law is consistent; with the increase in the thickness of the BST layer, the corresponding single-layer BST thickness at the peak of the output voltage of the structure is smaller. As the number of layers increases gradually, the weakening amplitude of the influence of the size effect on the output voltage of the structure after the thickness of the overall structure increases is smaller than that of the flexoelectric effect, and a sufficient number of superimposed layers is sufficient to compensate for the scale effect and the flexural effect. The energy loss caused by the electrical effect at this time, the thickness of the BST transducer layer can be appropriately increased when the laminated structure reaches the optimal output state, which can reduce the difficulty and loss of preparation in practical engineering applications to a certain extent.

Considering the difficulty of preparation of BST sensitive layer and the difficulty of attachment of too thin PDMS flexible layer in practical engineering applications, a BST sensitive layer with a thickness of 10 μm was selected as the basic element, and the PDMS layer was still 5 times the thickness of BST layer. The output voltage of the micro-scale flexural electric laminated beam energy harvesting device with different layers and thicknesses is calculated by the finite element method, as shown in Figure 8. It can be seen that with the increasing number of layers of microscale flexoelectric lamination beam under a constant thickness, the output voltage of the overall structure shows a linear growth trend, which proves that it is feasible to construct an energy harvesting device with both microscale effect and energy output characteristics with the thickness of BST and PDMS.

## 6. Conclusions

Based on the Lagrange multiplier method, the displacement and displacement gradient are constrained, and the C^1^ continuity of the element boundary is taken into account, a new mixed element is constructed. According to the parameter transformation method, a two-dimensional mixed finite element method related to the size effect and flexoelectric effect is developed.

By adjusting the proportion of BST layer and PDMS layer in the micro-scale flexoelectric laminated beam, the output voltage of the laminated beam structure with different structural sizes under static load was studied, and a three-layer micro-scale flexoelectric laminated beam structure was simulated to calculate the output voltage of the whole structure under uniform load, which was compared with the analytical solution and found that the error was very small. The effectiveness and universality of the finite element method are verified.

Finally, through the simulation of the multilayer microscale flexoelectric energy harvesting device, the influence trend of the scale effect and flexoelectric effect on the output voltage of the laminated beam in the multilayer structure state is compared, and the convenience of the finite element method can be seen by comparing with the theoretical analysis method.

## Figures and Tables

**Figure 1 micromachines-14-00755-f001:**
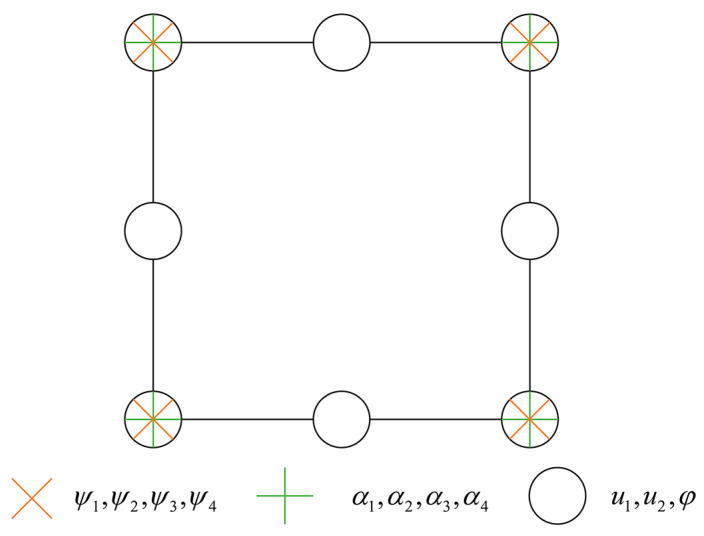
Two dimensional 8(4) Nodes flexoelectric mixed element.

**Figure 2 micromachines-14-00755-f002:**
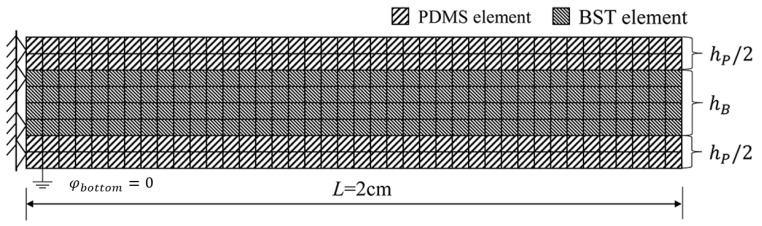
Finite element diagram of micro scale flexoelectric laminated beam.

**Figure 3 micromachines-14-00755-f003:**
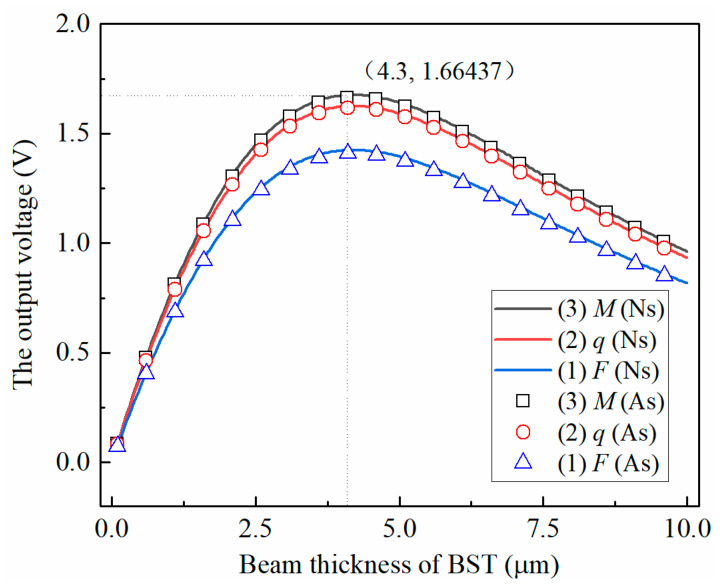
Voltage of the flexoelectric beam under the concentrated load *F*, distributed load *q*, and couple load *M* (Single layer BST) (Ns = Numerical solution, As = Analytical solution).

**Figure 4 micromachines-14-00755-f004:**
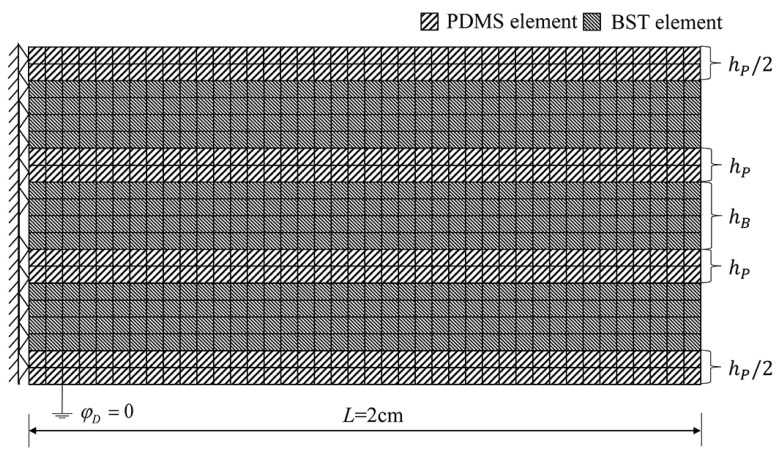
Schematic diagram of micro scale flexoelectric laminated energy collector.

**Figure 5 micromachines-14-00755-f005:**
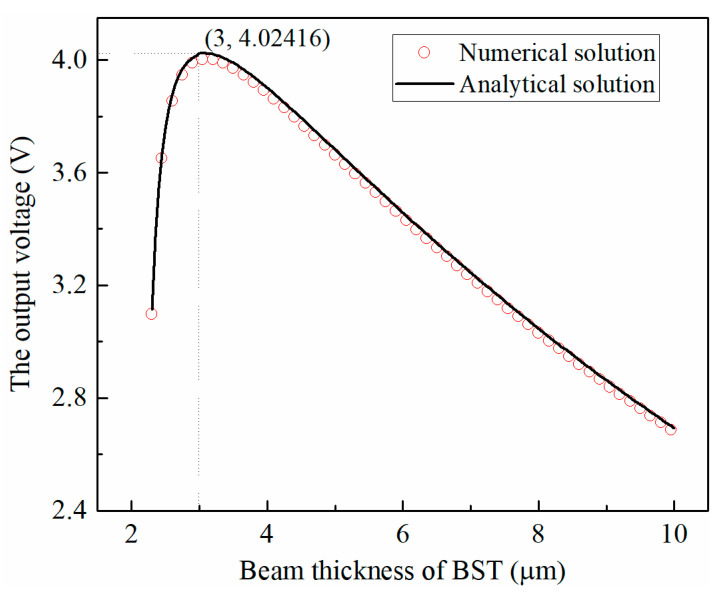
Output voltage of three-layer micro scale flexoelectric laminated beam under static load.

**Figure 6 micromachines-14-00755-f006:**
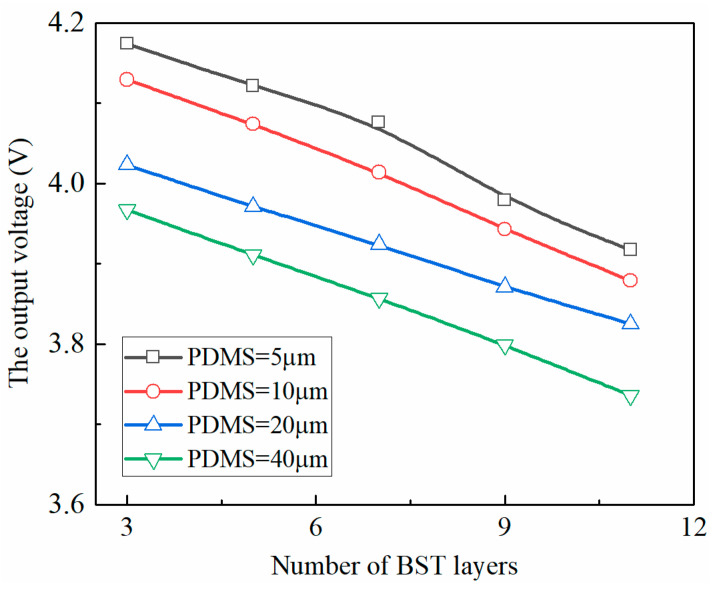
Optimal output voltage of multi-layer micro scale flexoelectric laminated beam energy collection device.

**Figure 7 micromachines-14-00755-f007:**
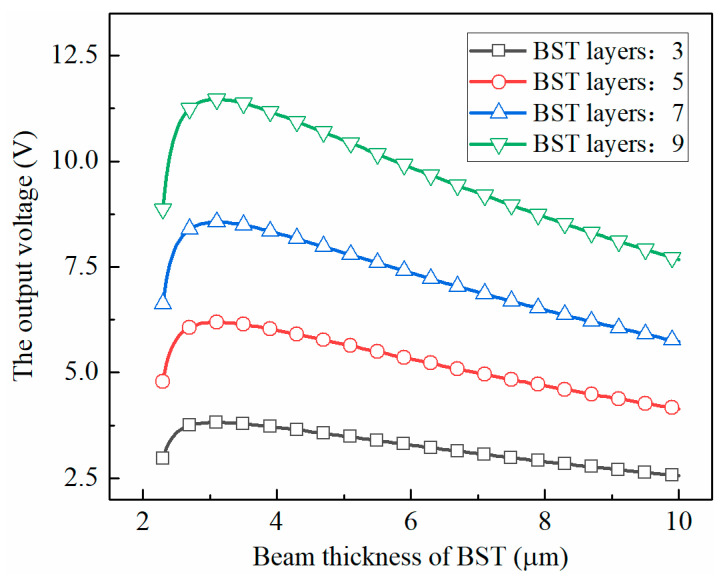
Output voltage of BST sensitive layer with different number of layers varying with the thickness of BST layer.

**Figure 8 micromachines-14-00755-f008:**
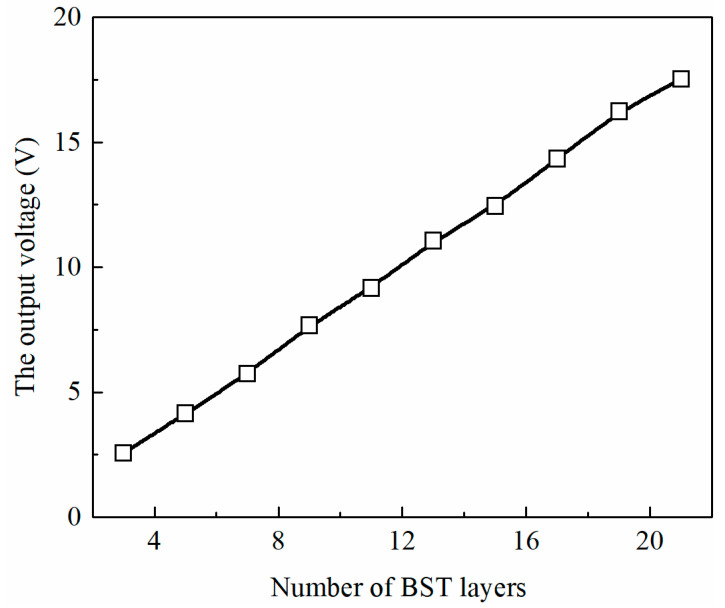
Output voltage of micro scale flexoelectric laminated beam energy collection device with fixed thickness.

**Table 1 micromachines-14-00755-t001:** Relevant material parameters of BST and PDMS.

	*E*	ν	ρ	$\mu_{14}$	$k_{33}$
BST	152 GPa	0.33	8.2 × 103 kg/m^3^	50 μC/m	13,200
PDMS	540 MPa	0.49	970 kg/m^3^		

## Data Availability

The data presented in this study are contained within the article. No additional data were created or analyzed in this study.

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
