# Peer review of "Research on Output Characteristics of Microscale BST Laminate Structure Based on Mixed Finite Element Method"

_micromachines, 2023, doi:10.3390/mi14040755_

Round 1

Reviewer 1 Report

This paper develops a mixed finite element method for estimating the output characteristics of microscale BST laminate structure. The technical work sounds good. The analytical solution can reach a good agreement with the numerical solution under different load conditions. The reviewer think this paper can be accepted after some minor revisions. And, there are a few minor issues that need to be addressed as follows:

(1)  Line 183: “On the basis of the previous paper, …”, Which paper is it ? The reviewer suggests an explicit citation of this paper.

(2) Line 199-201: There are actually three different load conditions. So, The reviewer suggests numbering the three load conditions, e.g. (1),(2),(3), which can providing a good reading an understanding.

(3) The size of these figures seem too small.

(4) Is the output voltage in Figure 3 an open circuit voltage? If this is the case, the reviewer recommends a clear statement.

(5) The references seem older. The reviewer suggests adding some new references.

(6) Reference [3] is in Chinese text. The reviewer suggests citing its English title.

Author Response

Dear:We are so appreciated for your letter on our manuscript (No: ISSN 2072-666X), entitled ‘Research on output characteristics of microscale BST laminate structure based on mixed finite element method’. We are also truly grateful to the referees’ critical comments and thoughtful suggestions on our manuscript. Based on these comments and suggestions, we have made careful modifications on the manuscript. And all changes made to the text are marked in red. We hope the new manuscript will meet your magazine's comments/ questions. If you have any other questions about this paper, we would quite appreciate it if you could let me know them in the earliest possible time.

Please see our point-to-point answers to the reviewers' comments/ questions below.

Thank you again for your time and consideration.

Sincerely yours,

Ying Luo

March 26, 2023

Below, the original comments are in black, and our answers are marked in red.

Additive list

To referee #1:

Comment 1

Line 183: “On the basis of the previous paper, …”, Which paper is it ? The reviewer suggests an explicit citation of this paper.

Answer

Thank you very much for your suggestion. Sorry for our inaccurate expression. What we want to express here is based on the previous section of this article, not the previous paper or work of other researchers.

The changes can be seen in line 183:“On the basis of the previous section of this article”.

Comment 2

Line 199-201: There are actually three different load conditions. So, The reviewer suggests numbering the three load conditions, e.g. (1),(2),(3), which can providing a good reading and understanding.

Answer

Thanks for your comments. We're sorry we didn't make it clear. As referee suggested that we have numbered the three loads in line 199-201, and made corresponding changes in Figure3.

Please see the Section 5.1 of our new manuscript.

Comment 3

The size of these figures seem too small.

Answer

Thanks for your suggestion and we have modified the size of all the figures.

Comment 4

Is the output voltage in Figure 3 an open circuit voltage? If this is the case, the reviewer recommends a clear statement.

Answer

Thank you for your comments. Yes, this is the open circuit voltage, we have not considered the reaction of the current to the flexoelectric effect during the process of turning on the circuit for the time being. And we have explained inline 208:” where the voltage represents the output voltage in the open-circuit environment”.

Comment 5

The references seem older. The reviewer suggests adding some new references.

Answer

Thanks for your suggestion. We apologize for the somewhat outdated selection of references and we have updated the references.

Comment 6

Reference [3] is in Chinese text. The reviewer suggests citing its English title.

Answer

Thank you very much for your comment. We are very sorry for our incorrect expression and the references have been revised.

Reviewer 2 Report

This paper presented a theoretical finite element method combining the size and flexoelectric effects

This paper presented a theoretical finite element method combining the size and flexoelectric effects with respect to the electromechanical coupling behavior of microscale flexoelectric materials. It introduced the displacement gradient as an essential parameter in addition to the displacement variable and formulated it as a parameter in a functional form. The introduction of the displacement gradient as an added parameter simplified the analysis of the electromechanical coupling behavior from the flexoelectric effect.

The structure of the paper is well arranged and, essentially, well written. Further editing may be helpful to improve the manuscript, especially the conclusion section, which is hard to read.

Overall, it is a nice theoretical paper. The manuscript can be more meaningful by comparing the results produced in this paper to some experimental measurements.

Author Response

Dear:We are so appreciated for your letter on our manuscript (No: ISSN 2072-666X), entitled ‘Research on output characteristics of microscale BST laminate structure based on mixed finite element method’. We are also truly grateful to the referees’ critical comments and thoughtful suggestions on our manuscript. Based on these comments and suggestions, we have made careful modifications on the manuscript. And all changes made to the text are marked in red. We hope the new manuscript will meet your magazine's comments/ questions. If you have any other questions about this paper, we would quite appreciate it if you could let me know them in the earliest possible time.

Please see our point-to-point answers to the reviewers' comments/ questions below.

Thank you again for your time and consideration.

Sincerely yours,

Ying Luo

March 26, 2023

Below, the original comments are in black, and our answers are marked in red.

To referee #2:

Comment 1

The structure of the paper is well arranged and, essentially, well written. Further editing may be helpful to improve the manuscript, especially the conclusion section, which is hard to read.

Answer

Thank you very much for your comment. We regret that the statement in the conclusion section is not clear enough, and we have modified this section:”

Based on the Lagrange multiplier method, the displacement and displacement gradient are constrained, and the C1 continuity of the element boundary is taken into account, a new mixed element is constructed. According to the parameter transformation method, a two-dimensional mixed finite element method related to the size effect and flexoelectric effect is developed.

By adjusting the proportion of BST layer and PDMS layer in the micro-scale flexoelectric laminated beam, the output voltage of the laminated beam structure with different structural sizes under static load was studied, and a three-layer micro-scale flexoelectric laminated beam structure was simulated to calculate the output voltage of the whole structure under uniform load, which was compared with the analytical solution and found that the error was very small. The effectiveness and universality of the finite element method are verified.

Finally, through the simulation of the multilayer microscale flexoelectric energy harvesting device, the influence trend of the scale effect and flexoelectric effect on the output voltage of the laminated beam in the multilayer structure state is compared, and the convenience of the finite element method can be seen by comparing with the theoretical analysis method.”
